# Factors Affecting Multimodal Transport during COVID-19: A Thai Service Provider Perspective

**Teerasak Charoennapharat** [1] and **Poti Chaopaisarn** [2,*]

1   Graduate Program in Industrial Engineering, Department of Industrial Engineering, Faculty of Engineering, Chiang Mai University, Chiang Mai 50200, Thailand; teerasak_ch@cmu.ac.th
2   Excellent Center in Logistics and Supply Chain Management, Faculty of Engineering, Chiang Mai University, Chiang Mai 50200, Thailand
*   Correspondence: poti@eng.cmu.ac.th

**Abstract:** Multimodal transport is a critical component in developing the international trade economy, and logistics service providers are a critical component in multimodal transport. However, the recent COVID-19 pandemic has seriously affected the transport system, especially in light of the ongoing rise in transportation costs which has increased firm operating costs. Furthermore, the COVID-19 pandemic has severely influenced the economic sector, resulting in decreased growth. This research aims to determine the priorities of the most important factors for developing and improving multimodal transport compared to pre-COVID-19 and during the COVID-19 outbreak. This research consisted of two stages. The first stage used bibliometric analysis to define multimodal transport dimensions and criteria based on the previous and current literature studies on multimodal transport and used confirmatory factor analysis (CFA) to verify the relationship between factors and multimodal transport. The second stage used the fuzzy best-worst method (FBW) combined benefit, opportunity, cost and risk (BOCR) to prioritize the improvement and development of multimodal transport during the COVID-19 crisis, which considers the perspective of logistics service providers in Thailand. These factors, when identified, would help policymakers design more efficient policies to improve and solve multimodal transport problems mainly caused by strict public health measures during COVID-19.

**Keywords:** COVID-19; multimodal transport; BOCR framework; factor analysis; FBW method; bibliometrics

## 1. Introduction

Multimodal transport plays a significant role in the global economy, especially in assuring the movement of goods from their place of origin to their destination. The concept has led to increased effectiveness of foreign and domestic trade, the development of industrial relations and the transportation of passengers and cargo [1]. However, the unforeseen pandemic event, COVID-19, has drastically impacted the operations of Multimodal Transport [2]. In order to reduce the number of infection cases, several governments closed or limited their trade borders, causing disruptions in vehicle movements, labor shortages and the preservation of physical distance in production facilities [3,4]. Due to the pandemic, several difficulties in multimodal transportation operations are expected to have a considerable impact on worldwide international trade [5]. For example, the goods transported in Colombia reached only 40% of the predicted values in the most critical months of 2020 [6]. In China, the impact of COVID-19 on multimodal transport is exemplified by the fact that long-haul trucking decreased to below 15% for the year 2019 alone, while in pre-pandemic times, it was expected to reach 92% by February [7] which is in accordance with the World Trade Organization's prediction that the global trade volume will decrease from 13% to 32% in 2021 [8].

The goal of a logistics and transportation provider is to provide services in the movement and storage of items using appropriate channels and tools [9]. However, Dwivedi and Hughes et al. [10] suggest that the COVID-19 epidemic has significantly and severely affected the existing methods of facilitating the flow of goods. It is worth noting that the pandemic revealed the vulnerability of business concerns and operations while also posing new obstacles. Due to travel limitations, some businesses were forced to temporarily close in order to remedy the outbreak. COVID-19 has created a challenge to force entrepreneurs to plan and adapt promptly in order to cope with these rapid changes. It is worth noting that this event is a global event that has a direct impact on the global economy. As referred to in the study by Apfalter, Hommes et al. [11], managing the supply chain across the borders and facilitating the means of trade have become some of the most challenging tasks in keeping a business afloat. As a result, logistics service providers have to respond to the changes while attempting to maintain operational performance and cost [12,13].

A significant decline in imports and exports, as well as supply chains issues, has impacted the performance of logistics service providers in Thailand [14]. Border trades in various Thai regions were impacted as a result of China's authorities closing borders between neighboring countries. Severe screening systems were put in place at the borders, which resulted in an increased delay in the export and import of commodities, such as agricultural goods, which are subject to safety procedures in order to boost confidence in trade [15].

However, in certain cases, providers of transport cannot solve the problem on their own and must seek government aid since some difficulties require government policies to be changed and enforced in order to control the problem [16]. Policymakers for COVID-19 have been confronted with the difficult challenge of helping transportation and logistics providers while staying within the economic and financial framework guidelines required to minimize economic impacts [16,17]. Therefore, the lack of guidelines for policymakers in making key judgments in times of crisis and inconsistent policymaking are influenced by inefficient technique development and a lack of appropriate information to support decisions. These issues can lead to significant economic losses [18].

Therefore, this study's main question is as follows: what are the critical factors involved in multimodal transport affected by the COVID-19 outbreak? From the perspective of logistics service providers in Thailand, what factors are most affected by the COVID-19 epidemic? These identified factors would help policymakers design more efficient policies to improve and solve multimodal transport problems mainly caused by the fallout of COVID-19.

The study process consists of three stages. The first stage employs bibliometric analysis to define multimodal transport dimensions and criteria based on literature studies, and the second stage employs CFA to verify the relationship between factors and multimodal transportation. The FBWM, based on the BOCR factor, was used in the final stage to rank the factors of multimodal transport affected by the COVID-19 crisis.

This research has contributed to assessing the economic conditions surrounding the current COVID-19 pandemic and its impact on the different factors of multimodal transport. We highlighted factors based on BOCR for a comprehensive perspective in the analysis of multimodal transport problems in an epidemic crisis. Currently, the literature developed on the impact of COVID-19, especially on the aspects of multimodal transport, is limited. Accordingly, it is imperative to research the post-pandemic economy since it has significantly affected the economic system and business sector [17,19]. Therefore, this investigation will contribute to important literature, especially when analyzing the impact of COVID-19 on multimodal transport.

## 2. Literature Background

### 2.1. Multimodal Transport

Multimodal transport refers to delivering products to clients via two or more modes [20]. Multimodal transport has become increasingly significant in the modern world because it

leads to a high rate of cross-border mobility at the regional level [21]. International multimodal transport refers to the movement of commodities from point A to point B under the control of a single transport operator [22]. Multimodal transport is like intermodal transport. It uses numerous modes of transportation to transfer goods from one location to another across international borders while using a single contract and single carrier [23].

Previous research involving multimodal transport has focused on multimodal transport route selection, for example, Banomyong and Beresford [20] examined various multimodal transport routes from Laos to the Netherlands with the inclusion of a confidence index for each route. Beresford, Pettit et al. [24] investigated the multimodal transport alternatives for iron ore from Northwest Australia to Northeast China, indicating the multimodal option for shipments of large bulk goods. Wang, Yeo et al. [22] examined the intermodal routing that optimized a road–rail intermodal transport network. Factors determined by the network's real-time condition, such as capacity, travel times, loading and unloading durations and container trains' fixed departure timings, are thought to be unpredictable in routing decision making. Sun [25] presents a study of intermodal routing for the modeling and optimization of a transportation routing problem in a road–rail multimodal transportation network that combines hub-and-spoke and point-to-point structures. Road transportation is time flexible, whilst rail transportation has set departure times, and the routing reliability is generated by simulating the uncertainty of the road–rail intermodal transportation network.

The majority of past multimodal route selection research has centered on building and merging multiple-criteria decision making (MCDM) [26]. For example, Meisel and Kopfer [27] used a mixed-integer programming approach to address the routing issue by synchronizing the path and evaluating the established model's performance. Singh, Gunasekaran et al. and Moslem, Ghorbanzadeh et al. [28,29] presented a Decision Support System (DSS) using Fuzzy-AHP for systemic analysis consisting of cost, traffic, risk factors, and DSS for the selection of the lowest cost, traffic and risk options. Mokhtari and Hasani [30] developed a goal programming model with chance constraints to choose the most influential international intermodal route, minimizing cost and risk while satisfying various on-time service requirements. Kaewfak, Huynh et al. [31] used risk analysis to model a fuzzy AHP and DEA, which was used to generate risk ratings for each route selection criteria. These methods are advantageous due to their simplicity, reasonability, comprehensibility, intuitive logic and ability to evaluate attributes using simple mathematical representations [32]. According to the findings of the aforementioned research studies, there are still limits in multimodal transportation research connected to the study and analysis of new aspects, which is necessary in order to adapt to the sustainability of each area [33]. Therefore, this research presents the study and analysis of factors affecting the development of multimodal transport in normal situations and during pandemics.

### 2.2. The Impact of COVID-19 on Multimodal Transport

COVID-19 has created complications and had adverse effects on the performance and profitability of most industrial and business sectors [19,34]. According to Loske [35], multimodal transport has been severely impacted by the outbreak of COVID-19 illness since the government has implemented various laws and limited firms' ability to conduct logistical operations. By the end of January 2020, the pandemic had been officially declared [36]. Governments in many nations enacted restrictions and rules to protect citizens and limit disease transmission [5]. However, according to Fernandes [37], these rules and policies have exacerbated the problems of the global economy by preventing enterprises from operating at their prior levels of growth (pre-2020). Additionally, the decline in commerce, manufacturing and other commercial sectors has adversely impacted transportation and logistics. Consequently, it has negatively influenced the total GDP growth rate in nations globally [36]. Furthermore, Aloi, Alonso et al. [38] investigated the impact of COVID-19 on urban mobility and found that total mobility fell by 76%, with public transport users falling

by up to 93%. Finally, trade barriers, demand restraints and a scarcity of trained labor have a substantial influence on supply chains and freight volume [35].

The intermodal transport of supplies for various products during the COVID-19 crisis had issues due to the restrictions on cross-border transport [39]. As more stringent public health policies lead to more regulations on cross-border transport, the cost and time of transportation increased [40]. This issue directly affects the business sector, including exporters, importers and logistics service providers. Regarding intermodal transport, efficiency has decreased, severely affecting the timely delivery of logistics and transportation services [41].

COVID-19 also has a direct impact on logistics providers. Logistics organizations, which are an essential part of value chains, help companies deliver their products to customers by facilitating trade and commerce within and across national borders. They were affected by pandemic restrictions, causing a reduction in competitiveness, economic growth and job creation [42,43]. The COVID-19 pandemic has exposed the vulnerability of operations in the provider sector and posed new problems. Due to the lack of coordination and collaboration, policymakers have had trouble assisting logistics service providers and exporters within the new governmental COVID-19 protocols [44]. Furthermore, organizations have sought to resolve the shift in consumer and supplier paradigms while also preserving potential operational and economic difficulties [34]; as such, the providers have had difficulty managing the logistics supply chain across borders and enabling commerce and business [45]. As a result, the important issues affecting freight and multimodal transportation were identified in this study to examine and investigate the key aspects of multimodal transportation that have been impacted by the COVID-19 pandemic on freight, particularly in the Thailand and South China trade corridor, and to identify key variables that would serve as a guideline for future epidemic management.

### 3. Methodology of the Study

#### 3.1. BOCR Analysis

BOCR analysis is similar to SWOT analysis, which considers not only a firm's strengths (S) but also its (external) opportunities (O) [46]. Opportunities in BOCR analysis often include criteria for positive spin-offs, future profits and income from future developments. At the same time, benefits reflect recent results in revenue or earnings [47]. Risks are designed to represent the expected implications of future events, whereas costs reflect (current) losses and efforts and the consequences of reasonably certain future developments. The BOCR model offers a potentially more profound study than benefit-cost (BC) analysis alone [48].

Therefore, the BOCR model was selected in combination with multi-criteria decision-making techniques to identify the important criteria of multimodal transport. BOCR can assist in establishing objectives or criteria to help decision-makers [49]. As a result, when compared to other related models, the BOCR model may include expert views in the decision-making process [50]. The BOCR model has been widely adopted in a variety of disciplines, providing a valuable resource for decision makers. For example, Liu, J., and Yin, Y. [51] utilized the model in sustainability projects through the performance assessment of energy storage nodes. Tsai et al. [52] proposed a two-stage model for evaluating the most suitable outsourcing logistics partnerships for a manufacturing factory. This research was used on FBWM–BOCR to prioritize important factors for alternatives to multimodal transport development to reduce the ambiguity of the perspective of LSPs and provide a more comprehensive overview of said factors.

#### 3.2. Fuzzy Best-Worst Method (FBWM)

The best-worst method (BWM) is a pairwise comparison-based strategy designed to handle MCDM issues [53]. BWM has two significant advantages over other MCDM methods: first, it requires less pairwise comparison data than a full pairwise comparison matrix; second, results generated by BWM are more consistent than those of other MCDM

methods (Ahmadi, Kusi-Sarpong et al. [54]) that use a full pairwise comparison matrix, which is also the primary reason for the use of BWM in this study. The approach has already been applied to several real-world challenges, for example, BWM has been employed to estimate the optimal arrangement for freight bundling while delivering freight from outstations to airports. Rezaei, Nispeling et al. [55] utilized the technique to pick the best suppliers based on environmental and economic parameters. Torabi, Giahi et al. [56] established a methodology for risk assessment in the scope of business management systems in order to evaluate the hazards detected.

However, given the complexity and unpredictability of objective objects and the fuzziness of human thought, using fuzzy information to represent decision information may be a more appropriate approach to various practical MCDM difficulties [57]. Meanwhile, other fuzzy-based MCDM approaches, such as fuzzy TOPSIS and fuzzy ELECTRE, have been presented and extensively utilized in recent years [58]. As can be seen, the MCDM approach is a very successful technique for resolving the assessment issue. According to the abovementioned study, combining multiple methods will provide more accurate and dependable experimental results. Therefore, this study used the fuzzy best-worst method (FBWM) to solve this problem. Additionally, the detailed steps of fuzzy BWM are as follows [59].

To determine relative weights, decision-makers were asked to make a pairwise comparison using a 1–9 scale. The pairwise comparison data were organized in the form of triangular fuzzy numbers:

1. Determine the best (most important) criterion and the worst (least important) criterion by B, O, C and R for each strategic factor. Based on the built decision criteria system, the best criterion and the worst criterion should be identified by LSPs in this step;
2. Execute the fuzzy reference comparisons for the best criterion. The fuzzy reference comparison is very important for FBWM;
3. Execute the fuzzy reference comparisons for the worst criterion. In this step, the other part of the fuzzy reference comparison is performed by using the linguistic evaluations of decision-makers listed in Table 1;
4. Solving the result by Equation (1).

$$\min \xi^* s.t. \begin{cases} \left| \dfrac{\left( l_B^w, m_B^w, u_B^w \right)}{\left( l_j^w, m_j^w, u_j^w \right)} - (lBj,\ mBj,\ uBj) \right| \leq (k^*,\ k^*,\ k^*) \\ \left| \dfrac{\left( l_j^w, m_j^w, u_j^w \right)}{\left( l_W^w, m_W^w, u_W^w \right)} - (ljW,\ mjW,\ ujW) \right| \leq (k^*,\ k^*,\ k^*) \\ \sum_{j=1}^{n} R(\tilde{w_j}) = 1 \\ l_j^w \leq m_j^w \leq u_j^w \\ l_j^w \geq 0 \\ j = 1,\ 2,\ \ldots,\ n \end{cases} \tag{1}$$

**Table 1.** Linguistic variables of decision makers adapted with permission from Ref. [60].

| Fuzzy Number | Linguistic | Fuzzy Number | | |
|---|---|---|---|---|
| 9 | Perfect | 8 | 9 | 10 |
| 8 | Absolute | 7 | 7 | 7 |
| 7 | Very good | 7 | 7 | 8 |
| 6 | Fairly good | 5 | 6 | 7 |
| 5 | Good | 4 | 5 | 6 |
| 4 | Preferable | 3 | 4 | 5 |
| 3 | Not bad | 2 | 3 | 4 |
| 2 | Weak advantage | 1 | 2 | 3 |
| 1 | Equal | 1 | 1 | 1 |

### 3.3. Research Structure

Based on the data from the literature research and expert local interviews for adaptation following local conditions, this study identified issues linked to the improvement of multimodal transport in cross-border areas and determined the factors that have the most critical effect on the development of multimodal transport in the Thai–South China Border Area. Figure 1 depicts the study phases.

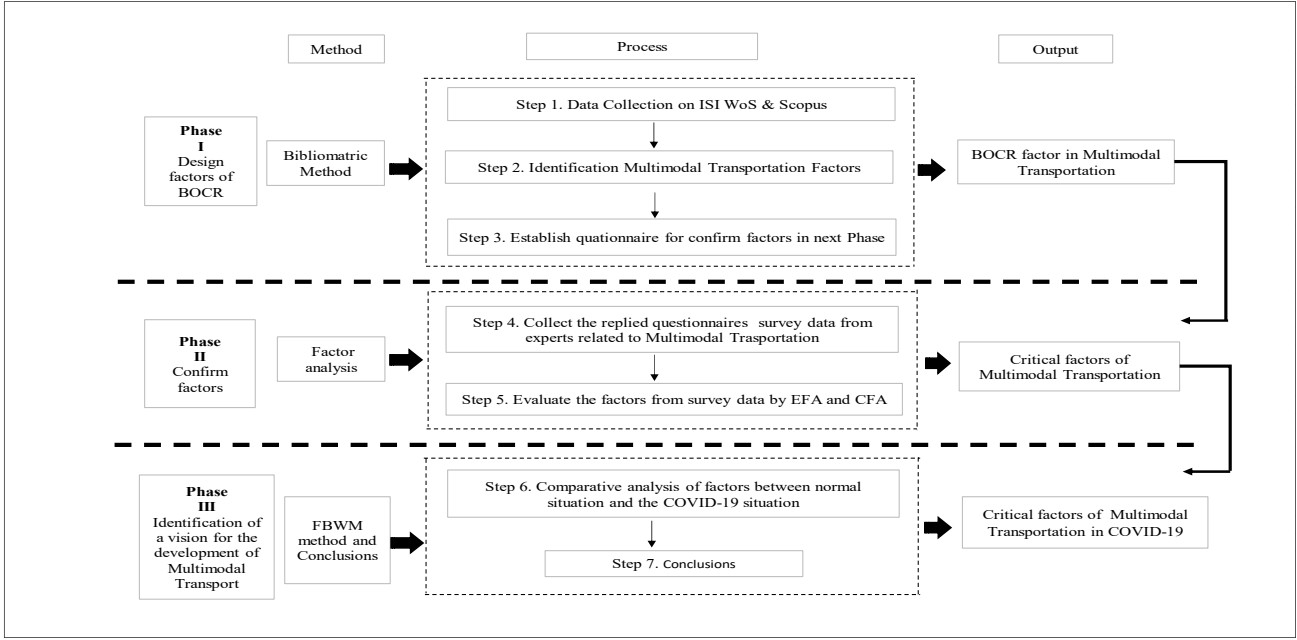

**Figure 1.** Research methodology.

### 3.3.1. Phase 1—Classifying Multimodal Transport Dimensions

The first step was classifying the critical dimensions in the route selection of multimodal transport. As mentioned above, multimodal transport dimensions were recovered from the literature review and data collection. The researcher gathered article titles and keywords from two web databases: Web of Science and Scopus index. The study period was 2010–2020, that is, from the year when widespread multimodal transport research began until the present. The database includes engineering, transport policy, economics, environmental science, expert systems with applications and related fields. Furthermore, this step refers to the dimensions from [61] the literature on multimodal transport, evaluation, model, criteria and related areas.

Table 2 shows the information that was used in a section of the identification factor by bibliometric analysis, which consisted of two online databases: Scopus index and Web of Science (ISI WoS), between 2010 and 2020. Furthermore, the database consists of the author's name, year, abstracts, keywords and address. The bibliometric method applies the abstract from the information. In this section, the abstracts were collected from the database using an abstract algorithm from the bibliometric Excel program, and information from the Scopus index and Web of Science were combined. This study focused on the dimensions for the development and selection route of multimodal transport implementation. The keywords used in the search were multimodal transport, intermodal transport, transport route selection, transport policy, multimodal transport development and important criteria in multimodal transport. Subsequently, this study examined journal articles and book abstracts to explore multimodal transport development and route selection.

**Table 2.** Bibliometric protocols.

| Information | Scopus, ISI Web of Science |
|---|---|
| Time Period | 2010–2020 |
| Classification | Transportation, Decision Sciences, Expert Systems with Applications, Location selection, Shipping and Logistics, Road transport, Logistic center, Multimodal Transportation |
| Source | Article; Book and Book Chapter |
| Keywords Investigated | Multimodal Transportation, Transportation Route Selection, Multimodal Transport Policy, Multimodal Transport Development, Benefit of Multimodal Transport, Opportunity of Multimodal Transport, Cost of Multimodal transport, Risk of Multimodal Transport |

Source: compiled by the authors.

Step 2. We identified multimodal transport dimensions. This study used bibliometric methods to evaluate the data from multimodal transport study databases and identify determination factors, the steps of which are shown in Figure 2. The steps are explained as follows: collected research from the Scopus database and the ISI Web of Science between 2010 and 2020; the condition consists of the author's name, year, abstracts, keywords, and address. After that, the research selection was made regarding the inclusion criteria referring to the keywords investigated in Table 2. Sections of research that do not fall within the inclusion criteria will be grouped into exclusion criteria and will be excluded in the next step.

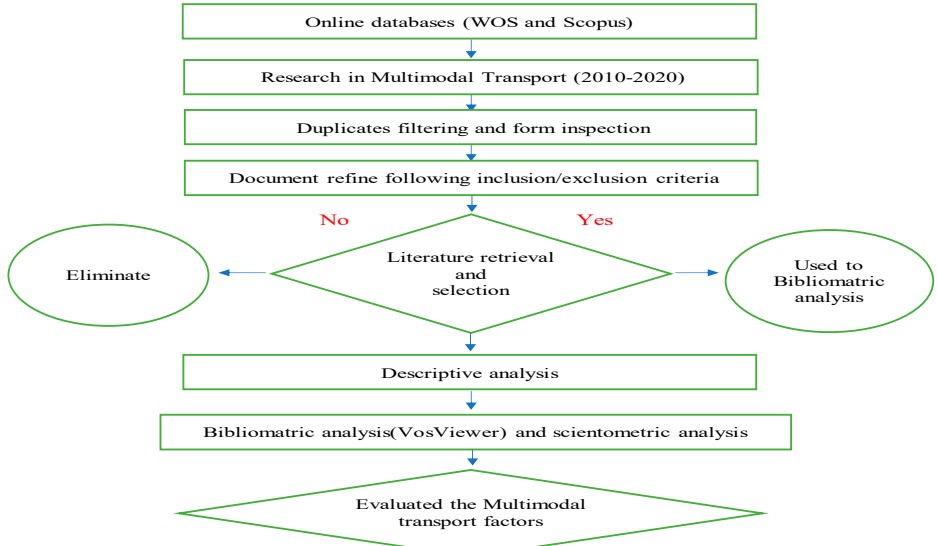

**Figure 2.** Multimodal transport dimensions identification using bibliometrics.

This method can evaluate study trends, which enhances the quality of database evaluation [62]. This can assist in decreasing the perspective bias resulting from researchers' prior research [63]. Consequently, bibliographic methods are quantitative techniques. Bibliometrics evaluated the data via a distance-based approach from the VOS Viewer software program, a simple-to-use demonstration software tool. This program concentrates on bibliographic networks [64]. The relations between keywords could be presented in a bibliography, which used the data of coupling, citation, co-citation, co-occurrence and co-authorship [65]. The results obtained from this step were used to create the questionnaire in the next step.

Step 3. We established a questionnaire for evaluating factors in the next phase. The target group was operators who had received registration from the Multimodal Transport

Operator of Thailand (MTO of Thailand), which had provided MTO services for at least five years. According to the Marine Department of Thailand (2016), it has 49 logistics operators that are small- and medium-sized entrepreneurs because COVID-19 impacted small and medium-sized entrepreneurs more than large firms [66]. The questionnaire was designed based on 52 factors divided into four dimensions (benefit, opportunity, cost and risk) derived from the previous step, with scores divided into 1–7 levels (1 = strongly disagree to 7 = strongly agree).

### 3.3.2. Phase 2—Evaluation and Confirmation of Multimodal Transport Factors

Step 4. Data collection from experts and LSPSs in multimodal transport amounted to 49 cases in Thailand through interviews and Google form questionnaires. During a four-month study, a total of 49 requests were made to all areas of Thailand, accompanied by an official letter from Chiang Mai University. The field survey was conducted by the author for this objective. Face-to-face and telephone interviews were the main information collection methods, and the questionnaire factors related to multimodal transport were analyzed using bibliometrics, as shown in Table 3. A total of 35 usable replies were obtained, reflecting a response rate of approximately 71%.

**Table 3.** Multimodal transport factors from occurrences and total link strength.

| No | Criteria | Occurrences | Total Link Strength |
|----|----------|-------------|---------------------|
| 1 | Signaling, Safety barrier | 18 | 57 |
| 2 | Warehousing | 10 | 37 |
| 3 | Customs clearance | 18 | 73 |
| 4 | Seasonal fluctuation of tariffs | 13 | 55 |
| 5 | City importance | 18 | 55 |
| 6 | Transport safety | 36 | 149 |
| 7 | Operating cost | 29 | 123 |
| 8 | Transportation cost | 13 | 46 |
| 9 | Freight damage rate | 13 | 50 |
| 10 | Maintenance cost | 14 | 57 |
| 11 | Insurance cost | 21 | 86 |
| 12 | Increase of market share | 48 | 253 |
| 13 | Tariff cost | 12 | 52 |
| 14 | Quality infrastructure | 10 | 144 |
| 15 | Stabilized relationship | 12 | 68 |
| 16 | Growth of International trade | 24 | 120 |
| 17 | Local political stability | 46 | 179 |
| 18 | Equipment utilization | 21 | 70 |
| 19 | International recognition | 19 | 85 |
| 20 | International agreement on FDI | 17 | 61 |
| 21 | Empty vehicle return rate | 30 | 83 |
| 22 | Excess of delivery time | 11 | 70 |
| 23 | Operational risk | 27 | 163 |
| 24 | Labor cost | 10 | 54 |
| 25 | On-time delivery ratio | 19 | 99 |
| 26 | Communication cost | 18 | 98 |
| 27 | Control section on the road | 18 | 69 |
| 28 | Development of high-tech | 39 | 155 |
| 29 | Government support | 14 | 67 |
| 30 | Collaboration gov and research institutions | 13 | 27 |
| 31 | Storage cost | 19 | 21 |
| 32 | Freight space availability | 146 | 175 |
| 33 | Time of delivery | 12 | 50 |
| 34 | Frequency of transportation | 44 | 168 |
| 35 | Mode connection efficiency | 14 | 47 |
| 36 | Load/unload cost | 13 | 39 |
| 37 | Category of road | 16 | 89 |

**Table 3.** *Cont.*

| No | Criteria | Occurrences | Total Link Strength |
|----|----------|-------------|---------------------|
| 38 | Risk of infrastructure and equipment | 26 | 140 |
| 39 | Collaboration with transport companies | 16 | 87 |
| 40 | City competitiveness | 18 | 96 |
| 41 | The impact of seasonality | 23 | 122 |
| 42 | Displacement | 11 | 48 |
| 43 | Parking lots | 29 | 163 |
| 44 | Petrol stations | 14 | 53 |
| 45 | Landscape | 11 | 40 |
| 46 | The weight of cargo | 27 | 134 |
| 47 | Distance of transport | 23 | 100 |
| 48 | Condition of the road surface | 19 | 58 |
| 49 | Forwarding partner | 14 | 46 |
| 50 | Juridical obstacles | 25 | 120 |
| 51 | Budget overrun | 53 | 243 |
| 52 | Impact of delay | 24 | 112 |

Step 5. The data were analyzed using the SPSS program for statistical analysis of each interviewee. Then, we analyzed dimensions and factors to measure reliability and validity because both are important in quantitative research [67]. Using the relationship between factors and mode by confirmatory factor analysis (CFA) as CFA is a critical technique in the complicated process of scale building. It allows for the assessment of an instrument or questionnaire's internal or latent structure [68]. CFA is often used in conjunction with exploratory factor analysis to confirm the presence of latent variables (factors or constructs) and the pattern of observed variable–factor correlations [68]. Thus, CFA can be quite beneficial in assisting researchers in determining how a factor should be graded [69]. For example, when an instrument's structure contains numerous latent variables, the pattern of observed variable–factor interactions revealed by a CFA enables the researcher to determine the number of factors employed as subscales in the research [70]. Therefore, the researcher tested the relationship dimensions and multimodal transport factors using CFA.

1.　Internal reliability refers to how well the measuring objects stay together when measuring a specific construct [71]. This reliability is achieved when the value of Cronbach's alpha exceeds 0.7.

2.　*KMO* is one of the indexes of factor analysis to check whether each factor is valid with the following equation:

$$KMO = \frac{\sum r_1^2}{\sum r_1^2 + \sum (partial\ correlation)^2} \tag{2}$$

According to the equation, if the value of *KMO* was less than 0.5, these data were not valid and were not considered in factor analysis. In the next stage, the relationship of each factor was analyzed using a correlation matrix [72].

3.　A correlation matrix with the following components was used [73]: correlation coefficients are displayed in a table called a correlation matrix; each table cell depicts the relationship between two variables; a correlation matrix was used to summarize data to enable more advanced analysis.

3.3.3. Phase 3—The Creation of BOCR Multimodal Transport Indicators

Step 6. We built the decision criteria system. The decision criteria system consisted of a set of decision criteria, which is especially important for evaluating alternatives. The decision criteria values can reflect different alternatives' performances to assess the problem and four dimensions of BOCR. Strategic factors were selected based on the characteristics of the research.

Final step. We determined the best (most important) and worst (least essential) criteria between multimodal transport in the typical scenario and COVID-19 using B, O, C and R. In this phase, LSPs should identify the FBWM based on the constructed decision criteria system.

## 4. Identification Multimodal Transport Dimensions and Factors by Bibliometric

The following section reviews multimodal transport through the bibliometric method, providing a more rigorous analysis of its method and trend. The result of the section is factors related to multimodal transport based on the BOCR model.

Bibliometric Analysis to Define Multimodal Transport Factors

This step discusses the approach for identifying data for bibliometric analysis. The bibliometric aided in identifying publishing patterns over time and authors, institutions, countries of origin, journals and keywords in the research area [74]. Simultaneously, bibliometric analysis is a common tool for identifying patterns and factors. Table 4 presents an advantage of the approach for identifying the factor or dimension.

**Table 4.** Comparison advantage of the method adapted with permission from Ref. [75].

| Approach | Process System | Reduces Ambiguity Bias | Traces the Aspects/Linkages | Allows the Hidden and Unexpected Aspects |
|---|---|---|---|---|
| Expertise | ✓ | | | |
| Questionnaire | ✓ | ✓ | | |
| Literature | ✓ | ✓ | | |
| Interview | ✓ | ✓ | | ✓ |
| Bibliometric | ✓ | ✓ | ✓ | ✓ |

As mentioned in the previous section, this study used a visualization from a similarity software application called VOS viewer [76,77]. Bibliometrics is a tool for collecting data that can be imported from various sources, for example, Scopus and Web of Science. It has been effectively used in many literature reviews [78–80], demonstrating its accuracy and reliability in science mapping. Therefore, this research used bibliometric analysis to define multimodal transport dimensions and criteria based on the earlier and present literature on multimodal transport, which was novel content in the multimodal transport dimension.

The following section explains the steps to analyze the dimensions of route selection and development in multimodal transport implementation. The bibliometric method is a popular method to analyze trends and factors. The bibliographic approach is a scientific analysis method, which can decrease experience bias from the expertise perspective, trace the relations between dimensions and expose hidden and unexpected information [81]. According to Table 3, the numbers of journals from searching keywords from Web of Science and Scopus were 620 and 885, respectively. From 1505 abstracts, the co-occurrence in the bibliometric method accounted for 336 keywords. After that, unrelated and duplicate keywords, such as route, decision and selection, were deleted. Hence, the investigator chose 52 criteria from the most common occurrences and total link strength in the existing literature review. The number of occurrences and total link strength are demonstrated [82]. In part, the total link strength describes the developments of individual keywords that encompassed all content in the resource. On the map, criteria are shown by circular badge symbols labeled on the sides, where dimensions are separated by different colors [65]. Therefore, the study showed 52 criteria. We updated the diagram to include a picture and the last 4-dimensional summary study for multimodal transport.

Figure 3 shows the link and clustering of dimensions derived from the visualization of similarities (VOS viewer) approach that builds maps via VOS viewer mapping [83]. The methods accomplished normalizing co-occurrence frequencies that examined the number of co-occurrences of node (criteria) A and node (criteria) B [84], which is called the relationship strength.

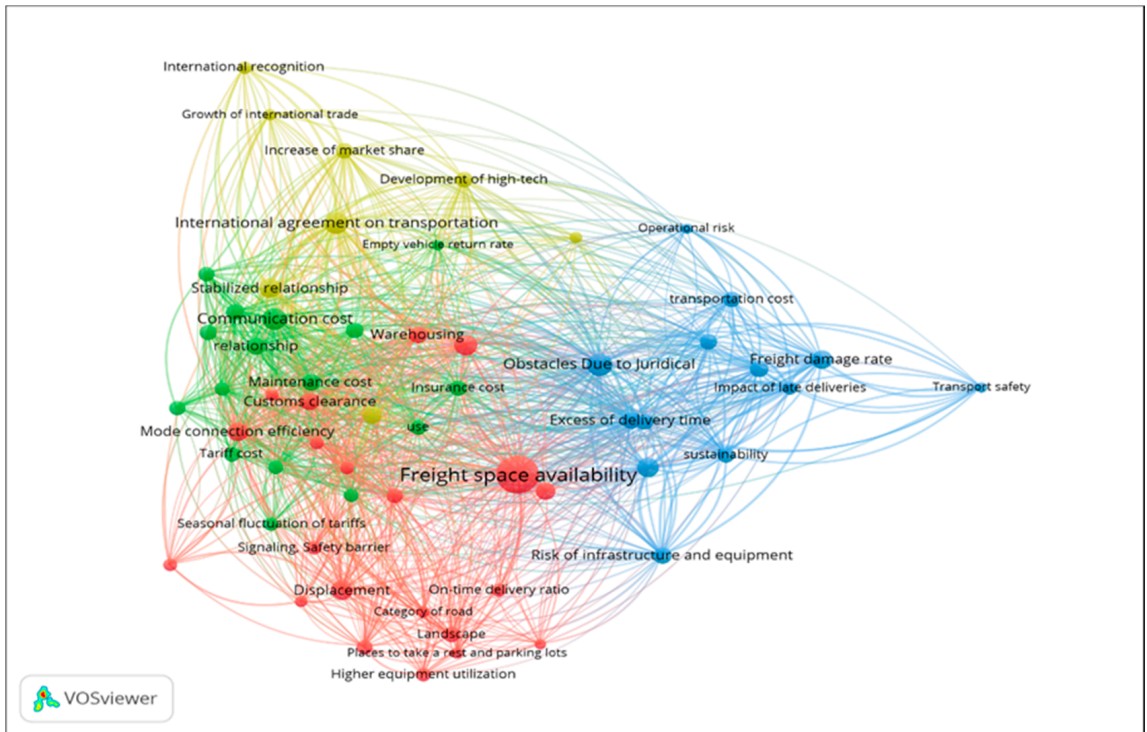

**Figure 3.** Multimodal transport correlation from visualization of relationships mapping.

Eck and Waltman, and Sun and Soergel [85,86] chose the factors from co-occurrence keywords and binary count from the VOS viewer techniques to gather the factors from the data. In later steps, the important keywords were collected by deleting the impertinent keywords, using four occurrences, 60% of the affinity term (default) and affinity scores of more than 0.4 [87]. In this step, multimodal transportation factors were categorized and determined in the dimension (pillars) through the visualization of similarity techniques in which each dimension is distinguished by a different color, where red is the benefit factor, green is the cost factor, yellow is the opportunity factor and blue is the risk factor.

Table 4 shows bibliometric program analysis results to determine which factors were most mentioned, thus leading the field of multimodal transport development. The top 5 factors with the highest score mentioned were transport cost, followed by budget overrun, increase in market share, local political stability and finally, transport safety. As the results show, the factors on which various research studies in multimodal transport focus are not just cost factors; in the next step, the types of factors were separated into four dimensions.

The four dimensions were interrelated, both within the same dimension and between them. The value indicating the link between factors was the total link strength. This higher value indicates that this factor was reasonably related to other factors. The total link strength with the most points was transport cost and budget overrun. These two factors had higher scores than the others because they are in the dimensions of cost and risk, which are typical aspects of transport analysis. Both factors could be referred to as fundamental factors before other factors will separate them in multimodal transport.

As shown in Table 5, all the dimensions related to transport were obtained and divided into four dimensions: benefits, opportunities, costs and risks. Typically, the dimension cost and risk are already widely used dimensions. However, in this research, two new dimensions were presented: benefit and opportunity.

**Table 5.** Multimodal transport dimension.

| Dimension | Criteria | Exemplary Publication |
|---|---|---|
| Benefits | (B1) Signaling, Safety barrier | [88] |
| | (B2) Warehousing | [89] |
| | (B3) Customs clearance | [90] |
| | (B4) City importance | [90] |
| | (B5) Quality infrastructure | [88] |
| | (B6) Local political stability | [91] |
| | (B7) Equipment utilization | [92] |
| | (B8) On-time delivery ratio | [92] |
| | (B9) Control sections on the road (police, cameras, carrier control) | [91] |
| | (B10) Freight space availability | [92] |
| | (B11) Frequency of transportation | [92] |
| | (B12) Mode connection efficiency | [92] |
| | (B13) Category of road | [88] |
| | (B14) Collaboration with transport companies | [93] |
| | (B15) City competitiveness | [92] |
| | (B16) Displacement | [94] |
| | (B17) Parking lots | [92] |
| | (B18) Petrol stations | [90] |
| | (B19) Landscape | [94] |
| | (B20) Distance of transport | [95] |
| | (B21) Forwarding partner | [92] |
| Opportunities | (O1) Increase in market share | [90] |
| | (O2) Stabilized relationship | [91] |
| | (O3) Growth of International trade | [96] |
| | (O4) International recognition | [97] |
| | (O5) International agreement on FDI | [98] |
| | (O6) Development of high-tech | [97] |
| | (O7) Government support | [99] |
| | (O8) Collaboration gov and research institutions | [97] |
| Costs | (C1) Seasonal fluctuation of tariffs | [96] |
| | (C2) Operating cost | [100] |
| | (C3) Transportation cost | [101] |
| | (C4) Maintenance cost | [101] |
| | (C5) Insurance cost | [101] |
| | (C6) Tariff cost | [96] |
| | (C7) Empty vehicle return rate | [92] |
| | (C8) Labor cost | [92] |
| | (C9) Storage cost | [92] |
| | (C10) Communication cost | [97] |
| | (C11) Load/unload cost | [22] |
| | (C12) Condition of the road surface | [90] |
| Risks | (R1) Transport safety | [101] |
| | (R2) Freight damage rate | [92] |
| | (R3) Excess of delivery time | [92] |
| | (R4) Operational risk | [101] |
| | (R5) Time of delivery | [22] |
| | (R6) Risk of infrastructure and equipment | [92] |
| | (R7) The impact of seasonality | [92] |
| | (R8) The weight of cargo | [101] |
| | (R9) Juridical obstacles | [97] |
| | (R10) Budget overrun | [96] |
| | (R11) Impact of delay | [22] |

## 5. Data Analysis Reliability and Validity Test

At this stage, each factor's reliability and validity were evaluated to be used in Step 6. In Step 5.1, Cronbach's alpha was checked to verify that the bibliometric program's

data were suitable for multimodal transport. Subsequently, in Step 5.2, the data from logistics service providers' questionnaires were analyzed to determine the relevant factors for multimodal transport development. In Step 5.3, Cronbach's alpha was checked again to confirm the suitability of dimensions and factors.

*5.1. Multimodal Transport Dimension Reliability Test*

This procedure explains the results of Cronbach's alpha analysis for each dimension. The scores analyzed were based on a rating of 35 experts. The research analyzed four dimensions: 21 factors for the benefit dimension, 8 factors for the opportunity dimension, 12 factors for the cost dimension and 11 factors for the risk dimension. After that, each dimension was examined to determine Cronbach's alpha. Table 6 shows the result of the data reliability based on Cronbach's alpha. The overall value of Cronbach's alpha of all four dimensions was 0.825: 0.743, 0.758, 0.885 and 0.840 for benefit, opportunity, cost and risk, respectively. Accordingly, the dimensions had high values of more than 0.7, indicating that the data were reliable. Consequently, the factors could be integrated into multimodal transport development indicators. The following section describes the synthesis of multimodal transport development indicators using questionnaires from the logistics service providers.

**Table 6.** Data reliability validation.

| Dimensions | Cronbach's Alpha | Number |
|:---:|:---:|:---:|
| Benefits | 0.743 | 21 |
| Opportunity | 0.758 | 8 |
| Costs | 0.885 | 12 |
| Risks | 0.840 | 11 |
| All dimensions | 0.825 | 52 |

*5.2. Multimodal Transport Dimensions and Factors Evaluated by Factor Analysis*

This procedure describes the data analysis results from the questionnaire interview conducted by a group of 35 logistics service providers in Thailand. All four dimensions: benefit, opportunity, cost and risk, were analyzed by the SPSS program. In this step, KMO and Bartlett's test values were used as criteria to indicate that the factors obtained in the previous step were related and appropriate to be used as overview factors in the development of multimodal transport. The Kaiser–Meyer–Olkin (KMO) of the BOCR dimension must be more than 0.7, and the significance level was less than 0.05.

As shown in Table 7, the KMO of the benefit variable was 0.869, more than 0.7, and significance = 0.000, which was less than 0.05. Therefore, the 21 variables were correlated enough to be analyzed by factor analysis. However, the benefit criteria had six factors. Those having extraction communality values of less than 0.5 were variables B1, B4, B15, B16, B19 and B21. Therefore, this research eliminated six factors. The conclusion was that 15 variables passed the analysis, as shown in Table 8.

**Table 7.** KMO and Bartlett's test of BOCR.

| Kaiser–Meyer–Olkin Measure of Sampling Adequacy | | Benefit | Opportunity | Cost | Risk |
|:---:|:---:|:---:|:---:|:---:|:---:|
| | | **0.869** | **0.713** | **0.742** | **0.743** |
| Bartlett's Test of Sphericity | Approx. Chi-Square | 313.015 | 258.255 | 278.453 | 331.866 |
| | df | 72 | 36 | 66 | 55 |
| | Sig. | 0.000 | 0.000 | 0.000 | 0.000 |

**Table 8.** Data reliability validation.

| | Factors | Extraction Communality Values | Cronbach's Alpha | Number |
|---|---|---|---|---|
| | All dimensions | | 0.882 | 36 |
| Benefit | | | 0.814 | 15 |
| B2 | Warehousing | 0.604 | | |
| B3 | Customs clearance | 0.601 | | |
| B5 | Quality infrastructure | 0.758 | | |
| B6 | Local political stability | 0.720 | | |
| B7 | Equipment utilization | 0.780 | | |
| B8 | On-time delivery ratio | 0.783 | | |
| B9 | Control sections on the road (police, cameras, carrier control) | 0.739 | | |
| B10 | Freight space availability | 0.573 | | |
| B11 | Frequency of transportation | 0.662 | | |
| B12 | Mode connection flexibility | 0.652 | | |
| B13 | Category of road | 0.731 | | |
| B14 | Collaboration with transport companies | 0.747 | | |
| B17 | Parking lots | 0.838 | | |
| B18 | Petrol stations | 0.664 | | |
| B20 | Distance of transport | 0.728 | | |
| Opportunity | | | 0.782 | 6 |
| O2 | Stabilized relationship | 0.781 | | |
| O3 | Growth of International trade | 0.637 | | |
| O4 | International recognition | 0.571 | | |
| O6 | Development of high-tech | 0.571 | | |
| O7 | Government support | 0.853 | | |
| O8 | Collaboration gov and research institutions | 0.520 | | |
| Cost | | | 0.902 | 8 |
| C2 | Operating cost | 0.687 | | |
| C3 | Transportation cost | 0.748 | | |
| C4 | Maintenance cost | 0.663 | | |
| C5 | Insurance cost | 0.606 | | |
| C7 | Empty vehicle return rate | 0.632 | | |
| C8 | Labor cost | 0.748 | | |
| C9 | Storage cost | 0.558 | | |
| C10 | Communication cost | 0.740 | | |
| Risk | | | 0.868 | 7 |
| R2 | Freight damage rate | 0.587 | | |
| R3 | Excess of delivery time | 0.523 | | |
| R5 | Time of delivery | 0.511 | | |
| R6 | Risk of infrastructure and equipment | 0.763 | | |
| R7 | The impact of seasonality | 0.728 | | |
| R10 | Budget overrun | 0.636 | | |
| R11 | Impact of delay | 0.622 | | |

As shown in Table 7, the KMO of the opportunity variable was 0.713, more than 0.7, and significance = 0.000, which was less than 0.05. Therefore, the 8 variables were correlated enough to be analyzed by factor analysis. However, the opportunity criteria had two factors. Those having extraction communality values of less than 0.5 were variables O1 and O5. Therefore, this research eliminated two factors. The conclusion was that 6 variables passed the analysis, as shown in Table 8.

As shown in Table 7, the KMO of the cost variable was 0.742, more than 0.7, and significance = 0.000, which was less than 0.05. Therefore, the 12 variables were correlated enough to be analyzed by factor analysis. However, the cost criteria had four factors. Those having extraction communality values of less than 0.5 were variables C1, C6, C11 and C12. Therefore, this research eliminated four factors. The conclusion was that 8 variables passed the analysis, as shown in Table 8.

As shown in Table 7, the KMO of the cost variable was 0.743, more than 0.7, and significance = 0.000, which was less than 0.05. Therefore, the 12 variables were correlated enough to be analyzed by factor analysis. However, the cost criteria had four factors. Those having extraction communality values of less than 0.5 were variables R1, R4, R8 and R9. Therefore, this research eliminated four factors. The conclusion was that 7 variables passed the analysis, as shown in Table 8.

*5.3. Scale Reliability*

Total factors concluded that 36 factors had passed the standards, divided into the benefit dimension, 15 factors; opportunity dimension, 6 factors; cost dimension, 8 factors; and risk dimension, 7 factors. The next step was to analyze Cronbach's alpha again to check after we eliminated factors. The study integrated the four dimensions and 36 factors into multimodal transport development to evaluate the new aspect and organized implementation to achieve the long-term capability and increase competitiveness. Table 8 shows the result of the data reliability of Cronbach's alpha. The overall 36 factors had a Cronbach's alpha value of 0.882, which was better than the first Cronbach's alpha before analysis via factor analysis, equal to 0.825. The dimensions of benefit, opportunity, cost and risk had Cronbach's alpha values of 0.814, 0.782, 0.902 and 0.868, respectively. Therefore, the measurements had high values of more than 0.7. This indicates the data were reliable. Therefore, the dimensions can be integrated into multimodal transport indicators. The following section presents the synthesis of multimodal transport indicators, concluding with multimodal transport indicators and definitions.

## 6. Scoring Multimodal Transportation Factors of Normal Situation and COVID-19 Situation by FBWM

After defining important indicators in multimodal transport for LSPs using the bibliometric method and factor analysis from the LSP perspective, this research obtained 36 essential factors in four dimensions to assess multimodal transport development. This paper addressed the score of multimodal transport factors of the normal situation and COVID-19 situation, using FBW/BOCR from the LSP perspective again via the interviews themselves. It aimed to explore multimodal transport factors affected by the COVID-19 crisis. This study mainly contributes to developing a guideline for solving multimodal transport problems during the pandemic now and in the future.

*Determining the Optimal Fuzzy Weights ($\tilde{w}^{*}1, \tilde{w}^{*}2, \cdots, \tilde{w}^{*}n$)*

A total of 36 factors, as shown in Table 8, were analyzed using FBWM following the equation from Section 3.2, in which the score at this stage was derived from the average of the scores from 35 logistics service providers in Thailand. Decision makers determined relative weights and were asked to make a pairwise comparison using the 1–9 scale in Table 2. The pairwise comparison data were organized in the form of fuzzy triangle numbers, with the result as shown in Table 9.

**Table 9.** The weight of multimodal transport dimensions and factors.

| | Factors | Weight Normally | Weight COVID-19 |
|---|---|---|---|
| Benefit | | 0.25 | 0.21 |
| B2 | Warehousing | 0.07 | 0.09 |
| B3 | Customs clearance | 0.1 | 0.12 |
| B5 | Quality infrastructure | 0.06 | 0.04 |
| B6 | Local political stability | 0.08 | 0.14 |
| B7 | Equipment utilization | 0.08 | 0.06 |
| B8 | On-time delivery ratio | 0.1 | 0.1 |
| B9 | Control sections on the road (police, cameras, carrier control) | 0.07 | 0.05 |
| B10 | Freight space availability | 0.05 | 0.04 |
| B11 | Frequency of transportation | 0.04 | 0.03 |
| B12 | Mode connection flexibility | 0.05 | 0.07 |
| B13 | Category of road | 0.03 | 0.03 |
| B14 | Collaboration with transport companies | 0.03 | 0.04 |
| B17 | Parking lots | 0.03 | 0.02 |
| B18 | Petrol stations | 0.03 | 0.03 |
| B20 | Distance of transport | 0.18 | 0.14 |
| Opportunity | | 0.2 | 0.18 |
| O2 | Stabilized relationship | 0.11 | 0.1 |
| O3 | Growth of International trade | 0.2 | 0.21 |
| O4 | International recognition | 0.2 | 0.19 |
| O6 | International agreement on FDI | 0.2 | 0.2 |
| O7 | Government support | 0.16 | 0.2 |
| O8 | Collaboration gov and research institutions | 0.13 | 0.1 |
| Cost | | 0.3 | 0.35 |
| C2 | Operating cost | 0.15 | 0.1 |
| C3 | Transportation cost | 0.36 | 0.46 |
| C4 | Maintenance cost | 0.06 | 0.03 |
| C5 | Insurance cost | 0.13 | 0.12 |
| C7 | Empty vehicle return rate | 0.1 | 0.1 |
| C8 | Labor cost | 0.08 | 0.07 |
| C9 | Storage cost | 0.06 | 0.08 |
| C10 | Communication cost | 0.06 | 0.04 |
| Risk | | 0.25 | 0.26 |
| R2 | Freight damage rate | 0.18 | 0.14 |
| R3 | Excess of delivery time | 0.15 | 0.15 |
| R5 | Time of delivery | 0.2 | 0.25 |
| R6 | Risk of infrastructure and equipment | 0.1 | 0.07 |
| R7 | The impact of seasonality | 0.13 | 0.12 |
| R10 | Budget overrun | 0.18 | 0.22 |
| R11 | Impact of delay | 0.06 | 0.05 |

Table 9 presents scores from the FBW method. The scores for various factors changed between the normal situation and the COVID-19 crisis from the LSP perspective in Thailand; it can be noted that the score for the positive dimensions (benefit and opportunity) was reduced, but the score for the negative dimensions (cost and risk) increased during the COVID-19 crisis. This corresponds to the study of Hobbs [39], which indicates that cost and risk factors became even more important during the pandemic. Which factors affected the changes in the score are explained in the next section.

## 7. Analysis of the Factors Affected by the COVID-19 Situation

This part ranks and compares each criterion's score from the previous section, as shown in Table 9, which addresses the multimodal transport factors of the normal and COVID-19 situations. It aims to explore multimodal transport factors affected by the pandemic. The result of this section will be used to develop guidelines for solving multimodal transport problems during the pandemic in the present and future.

### 7.1. Ranking Multimodal Transport Factors Based on Weight from FBWM

This step ranks the factors impacted by the COVID-19 outbreak, and it compares the percentage weight change for each factor between the pre-epidemic and epidemic crisis. The purpose of this process was to compare multimodal transport factor variations in the normal period to the COVID-19 period. Since the COVID-19 situation affected intermodal transport, this research compared multimodal transport factors during regular periods with the COVID-19 crisis to analyze which factors affected this epidemic state in order to establish a guideline for improving and solving real problems with logistics service providers. Table 10 shows the order of factors when sorted by weight obtained from the best-worst analysis, as well as shows the percentage change in each factor between the normal situation and the COVID-19 crisis. In terms of the normal situation, the first ten highest score factors were C3, R5, R2, R10, B20, C2, O3, O4, O5 and C5, whose weights were 0.108, 0.05, 0.045, 0.045, 0.045, 0.045, 0.04, 0.04, 0.04 and 0.039, respectively. The combination of the ten factors mentioned was 50% of the total weight. In terms of the COVID-19 period, the first ten highest score factors were C3, R5, R10, C5, R3, O3, O5, O7, R2 and C2, whose weights were 0.161, 0.065, 0.057, 0.042, 0.039, 0.038, 0.036, 0.036, 0.036 and 0.035, respectively. The combination of the ten factors mentioned was 55% of the total weight.

**Table 10.** Ranking the weight of multimodal transport factors.

| | Weight Normally | Ranking | Weight COVID-19 | Ranking | % of Change |
|---|---|---|---|---|---|
| Benefit | | | | | |
| B2 | 0.018 | 23 | 0.019 | 20 | 6% |
| B3 | 0.025 | 16 | 0.025 | 17 | 0% |
| B5 | 0.015 | 28 | 0.008 | 30 | −47% |
| B6 | 0.020 | 21 | 0.029 | 14 | 45% |
| B7 | 0.020 | 21 | 0.013 | 26 | −35% |
| B8 | 0.025 | 16 | 0.021 | 19 | −16% |
| B9 | 0.018 | 23 | 0.011 | 28 | −39% |
| B10 | 0.013 | 30 | 0.008 | 30 | −38% |
| B11 | 0.010 | 32 | 0.006 | 33 | −40% |
| B12 | 0.013 | 30 | 0.015 | 24 | 15% |
| B13 | 0.008 | 33 | 0.006 | 33 | −25% |
| B14 | 0.008 | 33 | 0.008 | 30 | 0% |
| B17 | 0.008 | 33 | 0.004 | 36 | −50% |
| B18 | 0.008 | 33 | 0.006 | 33 | −25% |
| B20 | 0.045 | 3 | 0.029 | 14 | −36% |
| Opportunity | | | | | |
| O2 | 0.022 | 20 | 0.018 | 21 | −18% |
| O3 | 0.040 | 7 | 0.038 | 6 | −5% |
| O4 | 0.040 | 7 | 0.034 | 12 | −15% |
| O6 | 0.040 | 7 | 0.036 | 7 | −10% |
| O7 | 0.032 | 13 | 0.036 | 7 | 13% |
| O8 | 0.026 | 15 | 0.018 | 21 | −31% |

**Table 10.** *Cont.*

|  | Weight Normally | Ranking | Weight COVID-19 | Ranking | % of Change |
|---|---|---|---|---|---|
| Cost |  |  |  |  |  |
| C2 | 0.045 | 3 | 0.035 | 10 | −22% |
| C3 | 0.108 | 1 | 0.161 | 1 | 49% |
| C4 | 0.018 | 23 | 0.011 | 28 | −39% |
| C5 | 0.039 | 10 | 0.042 | 4 | 8% |
| C7 | 0.030 | 14 | 0.035 | 10 | 17% |
| C8 | 0.024 | 19 | 0.025 | 17 | 4% |
| C9 | 0.018 | 23 | 0.028 | 16 | 56% |
| C10 | 0.018 | 23 | 0.014 | 25 | −22% |
| Risk |  |  |  |  |  |
| R2 | 0.045 | 3 | 0.036 | 7 | −20% |
| R3 | 0.038 | 11 | 0.039 | 5 | 3% |
| R5 | 0.050 | 2 | 0.065 | 2 | 30% |
| R6 | 0.025 | 16 | 0.018 | 21 | −28% |
| R7 | 0.033 | 12 | 0.031 | 13 | −6% |
| R10 | 0.045 | 3 | 0.057 | 3 | 27% |
| R11 | 0.015 | 28 | 0.013 | 26 | −13% |

*7.2. Comparative Analysis of Multimodal Transport Factors between Normal Situation and the COVID-19 Situation*

This procedure compared multimodal transport development factor weight changes during the normal situation and the COVID-19 crisis. As shown in Table 10, the study found that a total of 12 factors scored higher in the COVID-19 period. However, only seven factors were among the top 9 scoring factors, including C3, R5, R10, C5, R3, O7 and C7, each with a score of 0.161, 0.065, 0.057, 0.042, 0.039, 0.036 and 0.035, respectively. The sum of seven factors equaled 43.5% of the total score. The top five factors with the largest changes were C9, C3, B6, R5 and R10, which were 56%, 49%, 45%, 30% and 27%, respectively. This section explains why factors C3, R5, R10, C5, R3, O7 and C7 were important during the COVID-19 period. The information presented here was derived from additional interviews with the logistics service providers.

The most affected factor was factor transportation costs (C3), whose weight was 0.161, with an increase of 49% over the usual period because transport costs increased significantly during the COVID-19 crisis. As a result of regulations regarding cross-border cargo transport, new truck trailers or new trucks must be replaced by that country's vehicles. For example, trucks transporting goods from Thailand to China via land routes during normal situations can be used in Laos. Nevertheless, during the COVID-19 situation, the trucks had to be replaced by Lao vehicles. As a result, Lao logistics operators have increased their freight prices to 2–3 times regular prices because they have more negotiation power, implementing higher costs for Thai logistics operators. Operators rated factor C3 during the COVID-19 crisis as being increased by 49% because C3 is the most critical factor in transportation competition (Storeygard, Adam 2016). It is also a direct impact factor on the profit that the provider will receive. Hence, the greater the change in transportation costs, the greater the impact on logistics providers.

The second most rated factor was the time of delivery (R5), weighing 0.065, with an increase of 30% in weight. Due to more detailed procedures, such as cargo inspection and truck driver disease examination while passing through customs, the time of delivery was noted in the transport management. According to this factor, the service providers tended to select the routes with clear steps in order to reduce the time of goods transport. Consistent with research by Fang et al. (2020) [102], which examined the quality indicators of multimodal transport, in this study, it was indicated that time of delivery was beneficial for logistics service providers both in terms of easier management and reducing the risk of paying fines when transport has time constraints. Another study by Tsai. C (2021) [52]

indicates that the time of delivery of goods was an important factor in choosing a logistics provider during the COVID-19 crisis.

The third most rated factor was budget overrun (R10), weighing 0.057, a 27% increase; the rise in the score was due to the uncontrollable increase in costs due to COVID-19, for example, the increase in transport costs from the unannounced impacts of border closures forcing freighters to divert routes, leading to additional effects, such as more transit times, increased labor costs, additional employees to pay and fuel costs. The sub-factor related to R10 was the administrative cost of multimodal transport; operation costs accounted for 30.91% of multimodal transport research. Operation costs were those charges incurred in the daily running and were internal to the provider, consisting of fixed and variable costs [103]. Therefore, when R10 increases, the operation cost also increases.

The fourth most scored factor was insurance cost (C5), which was 0.042, an 8% increase. According to the weight, as shown in Table 11, this factor was significant in both the standard and COVID-19 situations. However, the score did not increase to a large extent because insurance costs are standard prices, not rapidly increasing or decreasing costs.

**Table 11.** Definition and indicator of critical factors of multimodal transport.

| Criteria | Definition | Example Indicator |
|---|---|---|
| B3 | Difficulty or ease of the process of sending goods through customs properly according to regulations. | Time spent in each custom. |
| B8 | Ratio of deliveries within the specified time period. | The percentage of goods delivered on time in each route. |
| B6 | The security of government or political in the area. | The frequency of policy changes or regulations regarding importing and exporting goods within one year. |
| B20 | The sum of the distances of all transportation stages. | Total distance required for transportation |
| O2 | The possibility of building trust and mutual reliance between LSP and LSP and service recipient to increase the competitive advantage of the supply chain. | The ratio of problems that occurred during the collaboration. |
| O3 | The growth rate of international trade is the exchange of goods and services between countries in the related area. | Percent of increase in the value of exports of goods passed through that custom compared to the past 5 years. |
| O4 | The level of reliability regarding the safety of that route. | Insurance price rates or incoterms used in the route. |
| O6 | The availability of high technology to support the transportation system to gain better effectiveness. | The level of technology in routes or border customs is used to facilitate the transportation of goods. |
| O7 | Government support such as policy, investment in infrastructure, and customs, in the route. | The route is in the government's current and future development plans. |
| O8 | The cooperation between business sectors and research institutions to provide. | The route has been studied and developed jointly between the private sector and educational institutions. |
| C2 | Expenses associated with the maintenance and administration of a business including the cost of transportation as well as overhead expenses. | Overhead cost, maintenance cost, communication cost. |
| C3 | Costs for transportation; costs for possible additional costs during transportation; additional insurance (insufficient safety). | The cost rate during the transportation of goods. |
| C5 | Insurance that covers the type of risks that are considered Marine, Aviation and Goods in International Transit (MAT) risks. | Insurance fee percentage (%). |

**Table 11.** *Cont.*

| Criteria | Definition | Example Indicator |
|---|---|---|
| R2 | The risk of freight damage is determined by the value of the damage and the amount of transport damage. It may be characterized as a circumstance in which items are lost during transport. | Percentage of damaged goods value. The situation of loss of products during transfer, damage from transportation, damage from delivery to customer in one period. |
| R3 | The postponement of delivering goods and services to customers. | The delivery rate is a delay when using the route. |
| R5 | Time for transportation; time for border crossing; time for customs clearance; exchange rate fluctuation during delivery time | Total time required for transportation. |
| R7 | Each season, weather with extreme conditions, such as overheating, extremely cold and pouring, leads to foggy vision and slippery roads. | The rate of weather effects that cause cargo damage or delays in transportation. |
| R10 | A cost increase which involves unexpected, incurred costs due to an underestimation of the actual cost during budgeting. | The value of the additional costs incurred during the carriage of goods on the route. |

The fifth highest score factor was excess of delivery time (R3), which was 0.039, an increase of only 3%. The score of R3 was not largely increased in the COVID-19 situation compared to the normal situation due to the fact that, in the opinion of the logistics providers, this factor was an essential factor in every situation, but it did not directly affect logistics service providers. However, the R3 factor is a priority factor for developing and solving current and future international shipping problems. R3 is an important factor affecting the customer's choice of logistics service provider [104]. In addition, during the COVID-19 crisis, R3 became increasingly important to compete with other providers [105].

The sixth factor was government support (O7) and had a score of 0.036, a 13% increase, which was a factor relevant to government policy, so it is more of a credibility issue during unusual situations such as COVID-19. Nevertheless, although providers pay attention, they regard some problems involving policy as difficult to solve by themselves during the COVID-19 crisis [105].

The last factor was C7, which scored 0.035, a 17% increase, which was the operator's cost dimension. The C7 factor was another operator of concern in both the regular situation and the COVID-19 period. However, the increased score was not directly attributable to the factor, but rather because the cost dimensions score was raised, as shown in Table 9. The researcher can conclude that the C7 factor from the provider perspective was as important as in the normal period, but C7 was not impacted by COVID-19. From the results of Table 10, this study can conclude that the most significant factors in the COVID-19 situation were cost and risk factors as they directly affected providers. However, B6 was another factor that saw a 45% increase in the score. Although the score was not among the top 10 highest scoring factors, a 45% increase in the score indicates the effect of COVID-19 on factor B6. The results from the instrumental analysis, consisting of bibliometrics, factor analysis and the fuzzy best-worst method, of the 18 key factors affecting multimodal transport during both normal and COVID-19 situations, are shown in Table 10.

**8. Conclusions**

In this research, we studied multimodal transport factors affected by COVID-19, as the pandemic was a new problem and lacked information to manage international transport; therefore, the authors identified the issues directly affecting the transport industry from surveys conducted by the Multimodal Transport Operator of Thailand (MTO of Thailand), which has been providing MTO services for at least five years. According to the Marine Department of Thailand in 2016, it had 49 operators. Regarding practical implications, this research highlighted the importance of the BOCR model from the literature and

the perspective of logistics service providers, which could provide a new perspective on conducting decisions related to multimodal transport operations, especially in the pandemic era. Reflecting on the findings of the research presented in Figure 1 and based on the research methodology illustrated in Figure 1, the result can be described as follows:

In Phase 1, this research presented the bibliographic approach as a scientific analysis method that reduces cognitive bias from expert experience, traces the dimension linkages and reveals the hidden and unexpected dimensions to define the BOCR dimension in multimodal transport, which can improve the analysis and development policy of multimodal transport. Concurrently, the BOCR model's effectiveness in the decision-making process is better than that of other relevant models [50].

In Phase 2, CFA was also used to help confirm the relationship between BOCR factors with multimodal transport. This was checked to verify that the bibliometric program's data were suitable for multimodal transport. The data were obtained through questionnaires from Thailand's logistics service providers' perspectives relevant to multimodal transport. Finally, this research used FBWM to prioritize the factors based on the BOCR model affecting the improvement and development of multimodal transport during the COVID-19 crisis to help policymakers and decision-makers develop solutions and support logistics service providers and the business sector of multimodal transport in a pandemic setting.

FBWM has two significant advantages over other MCDM methods: first, it requires fewer pairwise comparison data than a full pairwise comparison matrix; second, results generated by BWM are more consistent than those caused by other MCDM methods [54]. Therefore, FBWM can reduce the ambiguity of the perspective of LSPs and provide a more comprehensive view of factors.

BOCR analysis offers a potentially more profound and more comprehensive study than benefit–cost or cost-risk analysis, which is often used in multimodal transport applications. Therefore, the BOCR model was selected in combination with multi-criteria decision-making techniques for identifying the critical criteria of multimodal transport. BOCR can assist in establishing objectives or criteria to help decision makers make efficient choices. [49]. Moreover, the result of this research provides a new, comprehensive tool for assessing the current COVID-19 pandemic and its impact on the different factors of multimodal transport. Therefore, this research has two distinct contributions: an academic and a practical contribution. Regarding the academic contribution, this research wishes to contribute to the important literature, especially when analyzing the impact of COVID-19 and other pandemic crises on multimodal transport now and in the future. In terms of the practical contribution, this research wishes to use these factors to assist policymakers in developing policies to solve multimodal transportation difficulties during the COVID-19 crisis and other pandemic crises.

As a result of the research, it was identified that the affected factors in the COVID-19 situation were transportation cost (C3), time of delivery (R5), budget overrun (R10), insurance cost (C5), excess of delivery time (R3), government support (O7), empty vehicle return rate (C7) and local political stability (B6). These seven factors were essential in addressing transport problems during the pandemic. These helped policymakers design policies to improve and solve multimodal transport problems effectively with the COVID-19 situation. In response to COVID-19, policymakers may increase their support for improving logistics and the transportation sector's performance [7]. COVID-19 has had a detrimental effect on transportation and recovery from the pandemic is rather difficult. Therefore, governments can assist and support in improving performance and recovering from the post-pandemic situations of LSPs [41]. The transportation industry has made a substantial contribution to economic development; hence, the government must seek to enhance policy to recover the transportation industry from the LSP perspective to overcome the COVID-19 crisis. For some factors, providers cannot solve the problems by themselves and require assistance from the government because some issues must be improved by the policies that come from the government, which are enforced to control the problem. For example, controlling the freight price when entering another country requires cooperation between

governments. There must be a clear standard of controlling freight prices to not increase or decrease according to the carrier's needs. Accordingly, [14] the important factors for a logistics company in the COVID-19 crisis were studied, which found that standard cost influences competition in the transportation business.

**Author Contributions:** T.C.—conceptualization, methodology, data curation and analysis and original manuscript. P.C.—conceptualization, methodology, data analysis, administration, supervision and paper review. All authors have read and agreed to the published version of the manuscript.

**Funding:** This research received no external funding.

**Acknowledgments:** The author gratefully acknowledges respondents from the Excellence Centre in Logistics and Supply Chain Management of Chiang Mai University.

**Conflicts of Interest:** The authors declare no conflict of interest.

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
