# Peer review of "Factors Affecting Multimodal Transport during COVID-19: A Thai Service Provider Perspective"

_sustainability, doi:10.3390/su14084838_

Round 1
Reviewer 1 Report
This paper explored factors to multimodal transport and logistics from a service provider perspective in Thai. With the identified factors, this paper prioritized them to provide insights policymakers to minimize the impact of pandemics and improve the efficiency of logistics services. This paper is well organized and clearly delivers methodologies and findings. However, it should be further improved in terms of editing and clarity in how this study contributes to designing policies in Conclusion.
Author Response
Thank you for your comment,
please see the attachment
Kind regards, Mr. Teerasak Charoennaparat

Reviewer 2 Report
The paper attempted to propose a framework for multimodal transport. This study used a Fuzzy Best-Worst (FBW) approach combining Benefits, Opportunities, Costs, Risks (BOCR) to prioritize the most important factors for developing and improving multimodal transport compared to the pre-COVID-19 and COVID-16 periods. Clarifications and revisions are needed to fully appreciate the study. Please see the below comments.
1, In the introduction, it is suggested to give a short summary after line 69, which will be beneficial for the reader to get the main deliveries.
2, From line 84 to line 96, please add citations
3, While I appreciate the literature reported, its discussion is not well organized around the specific objective of this paper with the aim to clarify the choices. More importantly, some relevant and important literature is missing and should be included in the literature review.
For instance, first paragraph of section 2.1 should add following references
Emerging approaches applied to maritime transport research: Past and future. https://doi.org/https://doi.org/10.1016/j.commtr.2021.100011
Bi-level optimization model applications in managing air emissions from ships: A review
https://doi.org/https://doi.org/10.1016/j.commtr.2021.100020
Third paragraph of section 2.1 about route choice and multi-attribute decision making should add following references
Diverging effects of subjective prospect values of uncertain time and money https://doi.org/https://doi.org/10.1016/j.commtr.2021.100007
Cumulative prospect theory coupled with multi-attribute decision making for modeling travel behavior
Future transportation: Sustainability, complexity and individualization of choices
https://doi.org/https://doi.org/10.1016/j.commtr.2021.100010
4, “2.3. BOCR analysis”, “2.4. Fuzzy Best-Worst Method (FBW)”, and “2.5. Bibliometric analysis to define Multimodal Transport factors” are too long. Please streamline.
5, It is suggested to delate “3.1. Phase 1—Classifying Multimodal transport Dimensions”, “3.2. Phase 2—Evaluation and Confirmation of Multimodal transport Factors”, and “3.3 Phase3— The Creation of BOCR Multimodal Transport Indicators”.
6, “The overall value of Cronbach’s alpha of all four dimensions was 0.825, being 0.743, 0.758, 0.885, and 0.840 for Benefit, Opportunity, Cost, and Risk, respectively”. Is there any basis for the data here?
7, Please merge several sections of “5.2. Multimodal Transport Dimensions and Factors Evaluated by Factor Analysis.” and streamline it.
For example “5.2.1. Benefit Analysis”, “5.2.3. Cost Analysis” and “5.2.4. Risk Analysis” can be summarised as table.
8, Line 509 to line 522 should be placed at the beginning of the literature review.
9 In the conclusion section, it would be clearer if the conclusions were listed in separate sections.
10, The whole paper is really too long and it is suggested that some of the chapters be consolidated and simplified.
Author Response

(The authors gave the same response as above.)

Reviewer 3 Report
Dear Authors, you should keep investigating this important subject. I wish the comments that follow may contribute to your paper.

Author Response

(The authors gave the same response as above.)

Reviewer 4 Report
The structure of the paper itself is subordinated to the bibliometric analysis. The Materials and Methods section contains brief information on multimodal transport and the impact of covid 19 on international transport as well as the research methods applied. Materials and Methods in this paper is in fact partly a duplication and partly a slight extension of Introduction, which should not be the case.
Next, the authors have indicated the stages of the methodology. The order so adopted disturbs the reception of the paper. In the logic of the paper, the research problem, i.e. the impact of covid on multimodal transport, should be presented first and then the methodology, in which selected methods are discussed in more detail. Subsequent chapters depend on the methodology adopted. If the paper as a whole is a bibliometric analysis then the next chapter should be this analysis using the previously discussed methods and finally a discussion of the results and conclusions. If the second stage of the research is the authors' own research, conducted on the basis of the author's interview among logistics service providers, then the bibliometric analysis should be a separate chapter as the theoretical background.
In the structure presented by the authors, the problem itself, which is to be solved by the proposed methodology, is lost and the methods of text analysis, which are known and not the subject of the paper, come to the fore. The way in which references to literature sources are cited in the text also disturbs my perception of the text. I suggest changing the citation of footnotes in the body of the paper e.g. from " ref[18]" to "Beresford, A., S. Pettit, and Y. Liu".
At the same time, it is difficult to deduce from the text how the authors conducted the 2nd stage of the research concerning factor analysis. Did the authors conduct the interviews themselves? If so, how, on what sample, how the companies were selected, what was their structure (small large medium, carriers, forwarders, logistics operators?). Or did the factors result from the bibliometric analysis and the authors only prioritised them. If so, which interviews are involved? Definitely, in this paper it is worth making an illustrative drawing of the research procedure indicating what comprises a particular stage, what are its results, what are the assumptions and who performs it.
I consider the subject itself to be important and topical. Today we can already say that Covid 19 is an example of a disruption that has not been taken into account in previous risk analyses and is an example of the broad spectrum of the impact of a pandemic on commodity flows.
Author Response

(The authors gave the same response as above.)

Round 2
Reviewer 2 Report
Thank the authors' responses to my comments.
One minor comment is that there are many typos in the references.
Authors may use google scholar to obtain right reference citations.
Author Response
Please see the attachment
Kind regards,
Mr. Teerasak Charoennapharat

Reviewer 3 Report
Dear authors, it is noticeable that you made an effort to improve your article. Nevertheless, some aspects still need your attention. A thorough and detailed reading of your article seems already necessary. I send my comments in the annexed file.

Author Response

(The authors gave the same response as above.)

Reviewer 4 Report
The article is interesting and up-to-date. The introduced changes increased its quality. Congratulations to the authors.
Author Response
Kind regards,
Mr. Teerasak Charoennapharat